# Hyponatremia Associated with Prophylactic Low-Dose Trimethoprim during Systemic Corticosteroid Therapy for AQP4-Positive Optic Neuritis in a Diabetic Patient

**DOI:** 10.3390/antibiotics9040201

**Published:** 2020-04-23

**Authors:** Masahiro Takubo, Sho Tanaka, Masaru Kushimoto, Jin Ikeda, Katsuhiko Ogawa, Yutaka Suzuki, Masanori Abe, Hisamitsu Ishihara, Midori Fujishiro

**Affiliations:** 1Division of Diabetes and Metabolic Diseases, Department of Internal Medicine, Nihon University School of Medicine, Tokyo 173-8610, Japan; 2Department of Internal Medicine, Nihon University Hospital, Tokyo 101-8309, Japan; 3Division of Nephrology, Hypertension and Endocrinology, Department of Internal Medicine, Nihon University School of Medicine, Tokyo 173-8610, Japan; 4Division of Neurology, Department of Medicine, Nihon University School of Medicine, Tokyo 173-8610, Japan

**Keywords:** acidosis, aquaporin 4, hyperkalemia, hyponatremia, steroids, trimethoprim

## Abstract

Hyponatremia associated with low-dose trimethoprim in patients on concomitant systemic corticosteroid therapy has rarely been reported. Here, we describe a 57-year-old woman with a history of diabetes mellitus and hypertension treated with telmisartan, who presented with progressive visual impairment of the left eye due to anti-aquaporin-4 antibody-positive optic neuritis. The patient received pulsed intravenous methylprednisolone followed by oral prednisolone at 30 mg/day and trimethoprim–sulfamethoxazole prophylaxis (160 mg and 800 mg daily). Her serum sodium level steadily decreased, and the potassium level was slightly elevated despite well-preserved renal function. This state persisted even after telmisartan discontinuation. In addition to hypotonic hyponatremia (125 mEq/L) with natriuresis, hyperkalemic renal tubular acidosis was diagnosed based on normal anion gap metabolic acidosis and hyperkalemia with low urinary potassium excretion. After trimethoprim–sulfamethoxazole cessation, electrolytes and acid–base imbalances swiftly recovered. We can conclude that caution must be exercised when treating such patients, because even low-dose trimethoprim may cause hyponatremia concomitant with hyperkalemic renal tubular acidosis, despite the mineralocorticoid effects of systemic corticosteroids.

## 1. Introduction

Trimethoprim has broad-spectrum antibacterial activity by inhibiting dihydrofolate reductase to prevent microbial growth [1]. As a result of a synergistic effect, it is frequently used in combination with sulfamethoxazole for the treatment of respiratory, intestinal, cutaneous, and urinary tract infectious disease [1]. While trimethoprim–sulfamethoxazole is generally well-tolerated, its use is sometimes associated with diverse adverse effects on the neurologic, hematologic, cutaneous, reproductive, and renal systems [2]. Trimethoprim can elicit hyponatremia and hyperkalemic renal tubular acidosis (RTA) due to its structural similarity with amiloride [3,4,5]. Hyperkalemia is the most common manifestation of electrolyte/acid-base impairments, and it is potentially life-threatening [4]. Hyponatremia related to high-dose trimethoprim is also common, but low-dose trimethoprim rarely leads to hyponatremia in patients with preserved renal function [3,6]. The main countermeasure for these adverse events is the immediate discontinuation of trimethoprim and associated medications.

While the amiloride-like effect of trimethoprim is to inhibit epithelial sodium channels in the collecting tubule, mineralocorticoids positively regulate and increase renal sodium reabsorption; thus, patients with hyperkalemic RTA are sometimes treated with the mineralocorticoid agonist fludrocortisone [5]. A previous study has reported that concomitant corticosteroid therapy did not influence the incidence of hyponatremia associated with high-dose trimethoprim [3]. However, there is scant information on hyponatremia in patients using low-dose trimethoprim and concomitant corticosteroid. Herein, we describe a patient with no renal dysfunction who nonetheless exhibited hyponatremia related to prophylactic low-dose trimethoprim despite receiving systemic corticosteroid equivalent to a mineralocorticoid effect of 0.06 mg/day fludrocortisone.

## 2. Case Report

A 57-year-old woman with a history of aquaporin-4 (AQP4) antibody-positive optic neuritis presented with progressive visual impairment of the left eye over two days and was admitted for further evaluation and treatment. The patient had been diagnosed with anti-AQP4 antibody-positive optic neuritis two years earlier due to visual impairment of the right eye. However, no medical treatment had been initiated at that time due to the patient’s refusal. On watchful waiting, her visual symptoms had not deteriorated notably until her admission here. Her medical history also included hypertension and diabetes mellitus diagnosed at 47 years of age. The patient had developed diabetic polyneuropathy and proliferative diabetic retinopathy for which pan-retinal photocoagulation was given, but nephropathy had not emerged as a complication. Glycated hemoglobin on admission was 8.0%; at this time, medication consisted of 36 units of insulin glargine U-300 once-daily, anagliptin 200 mg/day, and metformin 500 mg/day. Hypertension was treated with telmisartan monotherapy at 40 mg/day alone and was satisfactorily controlled. The patient drank alcohol only socially and never smoked.

On physical examination, the patient’s body mass index was 20.9 kg/m^2^ (height 153 cm, weight 49.0 kg), body temperature was 36.1 °C, blood pressure was 107/55 mmHg, and pulse was regular at 86 beats/min. While Goldman perimeter testing revealed a newly developed central visual field defect in the left eye, the status of the diabetic retinopathy was not markedly changed. A bilaterally weakened Achilles tendon reflex and impaired vibration sensation over the medial malleolus was observed. Laboratory parameters on admission (Table 1) showed unremarkable electrolytes, preserved estimated glomerular filtration rate, and normo-albuminuria.

The clinical course of hospitalization is shown in Figure 1. From the fourth hospital day, the patient received pulsed intravenous methylprednisolone 1000 mg/day for three consecutive days, followed by oral prednisolone 30 mg/day. Additionally, low-dose trimethoprim–sulfamethoxazole prophylaxis (160 mg and 800 mg daily) was concomitantly administered. Subsequently, the serum sodium level steadily decreased, and the potassium level was slightly elevated despite normal renal function. A second course of pulsed methylprednisolone was given on days 13–15. Telmisartan was stopped on day 16, but electrolyte abnormalities did not improve. Therefore, on day 21, arterial blood gas analysis, urinalysis, and hormonal status tests were carried out in addition to routine examinations and body weight measurements (Table 2). Hypotonic hyponatremia with high urinary sodium excretion and mild hyperkalemia with low urinary potassium excretion were observed. While primary respiratory alkalosis was present, decreased bicarbonate levels greater than compensatory changes and negative base excess indicated the co-existence of primary metabolic acidosis. Based on the above findings and on the normal serum anion gap, the patient was diagnosed as having hyperkalemic RTA. After the discontinuation of trimethoprim–sulfamethoxazole, electrolytes and acid–base imbalances swiftly resolved. Visual impairment also improved and did not recur during prednisolone tapering, and the patient could be discharged on hospitalization day 29.

## 3. Discussion

Hyponatremia in hospitalized patients is a clinically important issue because it is associated with diverse adverse outcomes, including increased mortality [7]. The development of hyponatremia may be a side effect of many medications including diuretics, antidepressants, antipsychotics, antiepileptics, antidiabetics, nonsteroidal anti-inflammatory drugs, and anticancer agents [8]. Trimethoprim is pharmacologically similar to the potassium-sparing diuretic amiloride, and thus dose-dependently inhibits epithelial sodium channels in the renal-collecting tubule and causes increased sodium excretion and decreased potassium excretion [9,10]. Thus, trimethoprim use can lead to hyperkalemic RTA and hyponatremia.

Hyperkalemia is recognized as a common adverse reaction to trimethoprim, and its use at a high dose, as well as renal impairment, older age, and potassium-retaining medications are reported as contributory risk factors [4]. While hyponatremia frequently develops in patients treated with high-dose trimethoprim, it is less common in those exposed only to low doses of trimethoprim [4,6]. A recent retrospective study reported that the incidence of hyponatremia among hospitalized patients treated with high-dose trimethoprim was as high as 72.3%, with the severity of hyponatremia depending on the duration and cumulative dose of trimethoprim received [3]. On the other hand, another retrospective study reported that electrolyte disorders (hyperkalemia and/or hyponatremia) related to prophylactic low-dose trimethoprim were observed only in 17.5% patients with preserved renal function, and serum sodium levels were not significantly changed [6]. Furthermore, serum sodium concentrations were confirmed to be unchanged after low-dose trimethoprim in a different report, although serum potassium concentrations showed significant elevation in this prospective observational study [11]. In the present case reported here, hyponatremia and hyperkalemic RTA occurred during trimethoprim use at a prophylactic low dose of 160 mg/day and rapidly improved after its cessation. While this clinical course suggests an adverse event due to trimethoprim, several factors influencing the homeostasis of electrolytes and acid–base balance in the kidney may also have had a role in this case (Figure 2).

The first of these factors influencing homeostasis is the systemic corticosteroid that was given at the pharmacologic dose of 30 mg/day of prednisolone. Synthetic corticosteroids have drug-specific glucocorticoid and/or mineralocorticoid properties. The representative corticosteroid serving as a mineralocorticoid is fludrocortisone, which is useful for treating hyperkalemic RTA, primary aldosterone insufficiency, classic congenital adrenal hyperplasia, orthostatic hypotension, and septic shock [5,12,13,14]. Prednisolone is a frequently used corticosteroid for treating diverse inflammatory and autoimmune diseases because of its higher glucocorticoid activity relative to its mineralocorticoid activity. Nonetheless, prednisolone at a dose of 30 mg/day still provides a mineralocorticoid effect equivalent to 0.06 mg/day of fludrocortisone [12]. Thus, concomitant use of prednisolone can activate epithelial sodium channels in the collecting tubules and potentially compensate for the adverse effects of trimethoprim. However, our present case suggests that hyponatremia related to trimethoprim can still occur despite the use of systemic corticosteroid therapy and only a low dose of trimethoprim.

The second factor influencing homeostasis is the use of the angiotensin 2 receptor blocker (ARB) telmisartan in this patient. The co-prescription of trimethoprim and ARB is common in clinical practice due to the wide use of ARBs as anti-hypertensive drugs for patients with diabetes mellitus because of their beneficial renal protective effects [15,16]. However, in previous case-control studies, the use of trimethoprim–sulfamethoxazole together with renin–angiotensin–aldosterone blockade, including ARB, was reported to be associated with an increased risk of hyperkalemia-associated hospitalization and sudden death possibly due to hyperkalemia [17,18,19]. These results suggest that ARB can amplify adverse impacts of trimethoprim on the renal distal tubule. Although telmisartan was stopped on day 16 in the present case, high plasma renin activities and low aldosterone levels on day 21 and 25 indicated the residual inhibition of angiotensin 2-mediated aldosterone secretion in this patient. Therefore, telmisartan use could have contributed to electrolyte and acid–base imbalances in this case.

A third factor for consideration is the co-existence of diabetes mellitus in this patient. It is clear that patients with diabetes mellitus are susceptible to hyperkalemic RTA for several reasons, including low renin resulting from damage to the juxtaglomerular apparatus, autonomic dysfunction due to neuropathy, volume expansion due to renal salt retention, and weakened prorenin to renin conversion [20]. The patient´s baseline renin profile was not known due to the effects of treatment with trimethoprim, prednisolone, and telmisartan on the renin–angiotensin–aldosterone system, but it is possible that her diabetes mellitus was conducive to the development of hyperkalemic RTA.

In addition to an impaired renin–angiotensin system, free water retention can be related to the hyponatremia of this present case. Differential diagnoses of hypotonic hyponatremia with elevated urinary osmolality and high urinary sodium excretion include the syndrome of inappropriate antidiuretic hormone (SIADH), salt wasting, diuretic use, adrenal insufficiency, and vomiting [21]. Antidiuretic hormones mainly determine free water excretion by the kidney and play a crucial role in mechanisms responsible for the development of hyponatremia. While hypertonicity triggers antidiuretic hormone release via osmoreceptors, hypovolemia or a low effective plasma volume also leads to antidiuretic hormone secretion via baroreceptors [21]. Therefore, salt wasting due to trimethoprim results in volume depletion and raises antidiuretic hormone levels, which in turn results in free water retention [22].

Although body weight decrease during the hospitalization period indicated that salt wasting was the main mechanism at play here, SIADH is also pathophysiologically possible. Anti-AQP4 antibody is a disease-specific autoantibody in neuromyelitis optica (NMO) spectrum disorder, which is characterized by immune-mediated demyelination of the central nervous system predominantly affecting the optic nerves and spinal cord [23]. AQP4 is a protein widely expressed in the brain, skeletal muscle, lung, stomach, inner ear, and kidney, contributing to water homeostasis [24]. Although AQP4 is distributed throughout the central nervous system, the basolateral membrane of ependymal cells and glial lamellae of the supraoptic nucleus are particular localizations in the brain [24]. Corresponding to this characteristic localization of AQP4, the brain involvement of NMO is preferentially seen in the periventricular areas, including the hypothalamus [25]. Immune-mediated damage to the hypothalamus may lead to the dysregulation of antidiuretic hormone secretion. Indeed, recent reports indicate that SIADH can be concomitant with NMO, especially in patients with hypothalamic involvement [26,27]. Furthermore, SIADH can occur, albeit rarely, in patients without hypothalamic abnormalities on magnetic resonance imaging [28]. These findings indicate that SIADH can be a potential complication in any patient with NMO. SIADH remains a diagnosis by exclusion, and it is challenging to differentiate SIADH from salt wasting, because these two conditions show similar laboratory profiles [21]. Therefore, SIADH cannot be completely excluded in the case reported here. 

## 4. Conclusions

Hyponatremia concomitant with hyperkalemic RTA associated with trimethoprim can occur despite using only low doses of the drug, and even when combined with systemic corticosteroids with a mineralocorticoid effect. Diabetes mellitus and renin–angiotensin–aldosterone blockade may be exacerbating factors. Therefore, careful monitoring during trimethoprim therapy should be undertaken in patients with these risks.

## Figures and Tables

**Figure 1 antibiotics-09-00201-f001:**
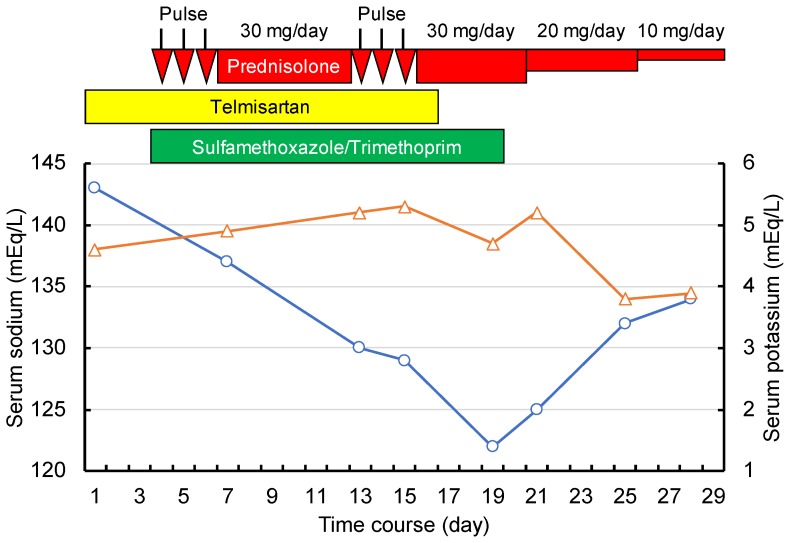
Clinical course. Horizontal axis indicates the time course. Vertical axes indicate electrolyte levels (blue circles and line, sodium level; orange triangles and line, potassium level). The red horizontal bar and arrows indicate steroid treatment. The yellow bar indicates telmisartan 40 mg/day. The green bar indicates trimethoprim 160 mg/day with sulfamethoxazole 800 mg/day.

**Figure 2 antibiotics-09-00201-f002:**
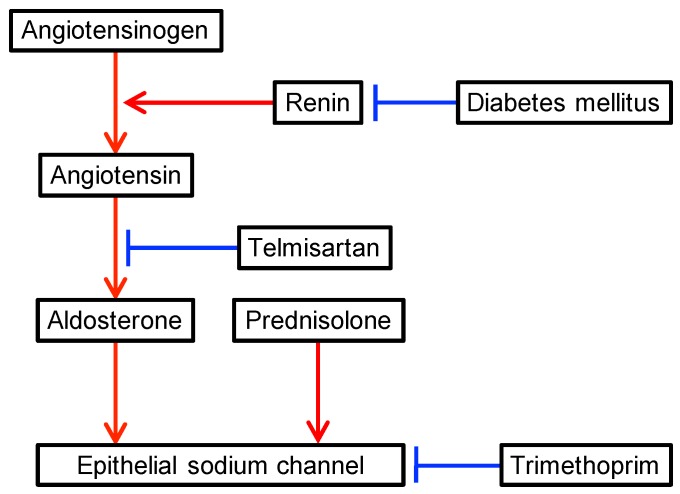
Factors influencing the renin–angiotensin–aldosterone system. Red and blue arrows indicate positive and negative regulatory effects, respectively.

**Table 1 antibiotics-09-00201-t001:** Laboratory parameters on admission.

Parameters	Values	Units	Reference Ranges
White blood cells	6600	/µL	(3300–8600)
Red blood cells	4.31	×10^6^/µL	(3.86–4.92)
Hemoglobin	13.2	g/dL	(11.6–14.8)
Platelets	133,000	/µL	(158,000–348,000)
Total bilirubin	0.68	mg/dL	(0.4–1.5)
Aspartate aminotransferase	18	U/L	(13–30)
Alanine aminotransferase	16	U/L	(7–23)
Gamma-glutamyl transpeptidase	17	U/L	(9–32)
Blood urea nitrogen	11.2	mg/dL	(8.0–20.0)
Creatinine	0.49	mg/dL	(0.46–0.79)
Estimated glomerular filtration rate	98	mL/min/1.73m^2^	/
Uric acid	3.8	mg/dL	(2.6–7.0)
Sodium	143	mEq/L	(138–145)
Potassium	4.6	mEq/L	(3.6–4.8)
Chloride	108	mEq/L	(101–108)
C-reactive protein	0.02	mg/dL	(≤0.2)
Total protein	7.8	g/dL	(6.6-8.1)
Glucose	81	mg/dL	(73–109)
Glycated hemoglobin	8.0	%	(4.6–6.2)
Thyroid stimulating hormone	0.84	µIU/mL	(0.34–3.8)
Free thyroxine	1.14	ng/dL	(0.8–1.5)
Anti-AQP4 antibody	≥40	U/mL	(<3)
Urine pH	5.5		(5–7)
Urine albumin	20	mg/day	/

Reference ranges are shown in parentheses. AQP4, aquaporin 4.

**Table 2 antibiotics-09-00201-t002:** Changes of clinical parameters just before and after trimethoprim cessation.

Parameters	Values			Units	Reference Ranges
	Day 21	Day 25	Day 28		
Body weight	45.4	44.3	43.8	kg	
Blood pressure	128/59	119/66	129/63	mmHg	
Pulse rate	81	83	89	beats/min	
Blood examination					
White blood cells	11,100	8300	4600	/µL	(3300–8600)
Red blood cells	4.15	3.62	3.83	×10^6^/µL	(3.86–4.92)
Hemoglobin	12.7	11.2	11.9	g/dL	(11.6–14.8)
Platelets	181,000	156,000	161,000	/µL	(158,000–348,000)
Total bilirubin	0.85	0.65	NM	mg/dL	(0.4–1.5)
Aspartate aminotransferase	13	23	13	U/L	(13–30)
Alanine aminotransferase	18	31	20	U/L	(7–23)
Gamma-glutamyl transpeptidase	19	20	13	U/L	(9–32)
Blood urea nitrogen	11.9	8.0	6.7	mg/dL	(8.0–20.0)
Creatinine	0.47	0.40	0.41	mg/dL	(0.46–0.79)
Estimated glomerular filtration rate	102.6	122.4	119.2	mL/min/1.73m^2^	
Uric acid	2.1	2.8	2.2	mg/dL	(2.6–7.0)
C-reactive protein	0.02	0.01	0.02	mg/dL	(≤0.2)
Total protein	5.9	5.3	5.7	g/dL	(6.6–8.1)
Glucose	149	96	113	mg/dL	(73–109)
Sodium	125	132	134	mEq/L	(138–145)
Potassium	5.2	3.8	3.9	mEq/L	(3.6–4.8)
Chloride	96	99	101	mEq/L	(101–108)
Osmolality	264	267	274	mOsm/kg H_2_O	(276–292)
Plasma renin activity	37	26	24	ng/mL/hr	(0.3–2.9)
Plasma aldosterone	43.2	38	167	pg/mL	(29.9–159)
Brain natriuretic peptide	7.7	NM	NM	pg/mL	(≤18.4)
Arterial blood gas analysis					
pH	7.438	7.465	7.469		(7.35–7.45)
CO2	26	32.3	30.4	mmHg	(35–45)
HCO_3_^-^	17.2	22.7	21.6	mEq/L	(22–26)
Base excess	−5.6	−0.4	−1.2	mEq/L	(–2–2)
Urine examination					
Sodium	92	46	137	mEq/L	(70–250)
Potassium	17.7	24.5	42	mEq/L	(25–100)
Chloride	79	47	151	mEq/L	(70–250)
Creatinine	43.8	58	111.3	mg/dL	(100–150)
Osmolality	510	614	785	mOsm/kg H_2_O	

Reference ranges are shown in parentheses. NM, not measured.

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
