# Peer review of "Hyponatremia Associated with Prophylactic Low-Dose Trimethoprim during Systemic Corticosteroid Therapy for AQP4-Positive Optic Neuritis in a Diabetic Patient"

_antibiotics, 2020, doi:10.3390/antibiotics9040201_

Round 1

Reviewer 1 Report

Good manuscript by Takubo and coworkers. The premise is interesting, the goal challenging and the results attractive. Overall the paper contains valuable elements and may be publishable after addressing some points.

In this case report Takubo et al  presents describe a 57-year-old woman with a history of diabetes mellitus and hypertension treated with telmisartan, who presented with  progressive visual impairment of the left eye due to anti-aquaporin-4 antibody-positive optic neuritis.

Authors conclude that treating such patients because even low-dose trimethoprim may cause hyponatremia concomitant with hyperkalemic renal tubular acidosis despite the mineralocorticoid effects of systemic corticosteroids.

I only have three minor observations before recommending publication in Antibiotics:

  1. Introduction: I think that the authors should give the readers more information about trimethoprim: use in therapy, side effects. The authors should refer to the latest publications in this area [Current Medicinal Chemistry (2019) 26: 1; J Antibiot 73, 5-27 (2020)].
  2. In my opinion, the authors should include information on laboratory parameters after trimetoprim treatment for comparison (respectively to Table 1).
  3. In laboratory parameters not include red blood cells. The parameter is important. Application of the preparation to persons deficient in glucose 6-phosphate dehydrogenase may lead to the risk of hemolysis (irreversible blood cell damage). In case of application of the preparation in these individuals, minimum doses of the drug should be applied and caution should be exercised (observe whether there are no symptoms of hemolysis by performing appropriate laboratory tests).

Reviewer 2 Report

It has been reported and seen before TMP/SMX in addition to acting as a K sparing diuretic is also going to impair free water clearance resulting in hypoNa

Reviewer 3 Report

In this study (Takubo M. et al), the authors report on a case of hyponatremia that they attribute to trimethoprim-sulfamethoxazole administration in a diabetic patient on concurrent corticosteroid therapy.  It is important to note that this patient has anti-aquaporin-4 antibodies that are the cause of her optic neuritis, and that patient had documented elevated levels of these antibodies on admission as shown in Table1.

The authors provide a good review of the known mechanistic effects on trimethoprim on the renal tubular sodium.  The authors demonstrate hyponatremic and hyperkalemic findings that appear to correct with cessation of this antibiotic and so conclude that the hyponatremia, which is less commonly seen with low-doses of this medication, warrants publication as a case report.

I have concerns that a potential role for concurrent syndrome of inapporiate antidiuretic hormone (SIADH) as a contributor and/or risk factor for hyponatremia in this patient is not addressed.  The volume status of this patient is not mentioned in the article which would be important to determine if SIADH may play a role.  Aquaporin-4 (AQP4) is expressed in the inner medulla of the kidney and is constitutively expressed in the basolateral cell membrane of principal collecting duct cells to provide a pathway for water regulation.  SIADH leading to hyponatremia has in fact been reported in patients with circulating anti-AQP4 antibiodies (ie. Nakajima H. et al. Case Rep Neurol. 2011 Sep-Dec; 3(3): 263–267); Inoue K. et al. BMJ Case Rep. 2017 Apr 20. doi: 10.1136/bcr-2017-219721).  Therefore I do not believe that hyponatremia in this patient could be determined to be fully attributable to the antibiotic alone.  The elevated urine osmolality, decreased serum osmolality, and elevated urine sodium concentration in the face of hyponatremia in this patient suggests this possibility.

This case report is otherwise well written, but is missing this potential contributor, as well as some key clinical features of the report patient such as volume status, which make it difficult to conclude that the low-dose trimethoprim-sulfamethoxazole could be the sole cause of this patient’s hyponatremia.

Round 2

Reviewer 3 Report

In this latest (v2) revision of the study by Takubo M. et al, the authors have addressed initial concerns raised regarding possible concurrent SIADH and have discussed the difficulty of fully differentiating SIADH from sodium wasting.  The authors have also provided a more comprehensive discussion around SIADH and AQ4.  The conclusions brought forward by the authors now appear appropriate for the results, and the study remains valuable and interesting for readers given that hyponatremia is less-widely reported with low-dose (as opposed to high-dose) trimethoprim-sulfamethoxazole.  I believe this manuscript is now acceptable for publication.